# Measuring of the COVID-19 Based on Time-Geography

**DOI:** 10.3390/ijerph181910313

**Published:** 2021-09-30

**Authors:** Zhangcai Yin, Wei Huang, Shen Ying, Panli Tang, Ziqiang Kang, Kuan Huang

**Affiliations:** 1School of Resources and Environmental Engineering, Wuhan University of Technology, Wuhan 430070, China; yinzhangcai@whut.edu.cn (Z.Y.); tpl@whut.edu.cn (P.T.); ziqiangkang@whut.edu.cn (Z.K.); 253463@whut.edu.cn (K.H.); 2School of Resource and Environmental Sciences, Wuhan University, Wuhan 430070, China; shy@whu.edu.cn

**Keywords:** time geography, COVID-19, space-time prism, space-time path

## Abstract

At the end of 2019, the COVID-19 pandemic began to emerge on a global scale, including China, and left deep traces on all societies. The spread of this virus shows remarkable temporal and spatial characteristics. Therefore, analyzing and visualizing the characteristics of the COVID-19 pandemic are relevant to the current pressing need and have realistic significance. In this article, we constructed a new model based on time-geography to analyze the movement pattern of COVID-19 in Hebei Province. The results show that as time changed COVID-19 presented an obvious dynamic distribution in space. It gradually migrated from the southwest region of Hebei Province to the northeast region. The factors affecting the moving patterns may be the migration and flow of population between and within the province, the economic development level and the development of road traffic of each city. It can be divided into three stages in terms of time. The first stage is the gradual spread of the epidemic, the second is the full spread of the epidemic, and the third is the time and again of the epidemic. Finally, we can verify the accuracy of the model through the standard deviation ellipse and location entropy.

## 1. Introduction

At the end of 2019, the COVID-19 pandemic has become a public health event affecting the whole world. It has left deep traces on the political activities, economic activities and people’s daily lives in many countries and regions. It has profoundly affected the spatial mobility of many populations. National leaders used video conference systems to communicate with government officials. Workers were unable to commute and some of the factories could not work as usual, which lead to a spate of factory closures and layoffs. Service industries implemented a strict daily appointment system and limited the number of people. People needed to be medically isolated if they were identified as confirmed patients or close contacts. The spread of this virus shows remarkable temporal and spatial characteristics. In the history of GIScience, there has never been a time when movement analytics and mobility insights have played such an important role in decision-making as in today’s global response to the COVID-19 crisis [1]. More than ever, today, we recognize the importance of mobile data and mobile analytics in crisis mitigation and public health [2].

Geospatial disparities have a great influence on the temporal and spatial distribution of the COVID-19 pandemic. In terms of regional size (regions, countries or even a continent), because they have different geographic spaces, the temporal and spatial distribution of the COVID-19 pandemic will show different laws [3,4]. Moreover not only will the size of the area affect the temporal and spatial distribution of COVID-19, but also factors such as health care capacity, population mobility, topography, and economic level will also lead to different temporal and spatial distributions of the disease. There are also many ways to explore the temporal and spatial distribution of COVID-19, such as exploratory spatial analysis methods, Poisson distribution, IDW and Kriging interpolation [5,6]. These scholars consider COVID-19 as a whole object and conduct research on the spatial distribution and temporal changes of the pandemic from the perspective of GIS. Due to different factors such as geographic environment and policy effectiveness, the distribution of COVID-19 presents quite a lot of heterogeneity in locational and temporal preferences. They described and visualized COVID-19 from two dimensions of time and space, but they did not consider the constraints of time and space. Time geography provides a theoretical basis for the modeling of space-time constraints too.

Time geography is an important tool for studying space-time uncertainty based on space-time constraints. It converts space-time constraints into space-time paths and space-time prisms to describe space-time uncertainty. Movement analytics have become an important part of many disciplines, including geographic information, sports ecology, transportation, and public health [7]. Time geography is an important method of exploring moving objects, in which we consider the COVID-19 pandemic as a moving object. In fact, time geography has begun to be employed by some scholars in the study of COVID-19. The main content of their research is to show the temporal and spatial indeterminate distribution of COVID-19 based on the accurate movement trajectory of confirmed patients. The concept of constraints in time geography is an important point in this theory. Due to the highly contagious nature of the infection, almost all countries and regions have adopted many restrictive measures to protect the lives and property of their residents. These restrictive measures have the same meaning as the three constraints in time geography to a certain extent [8,9]. Similarly, two important tools in time geography—space-time path and space-time prism—can also be used to simulate the spread of the COVID-19 pandemic [10,11,12,13]. The time-space path is often used to simulate the life trajectory of the coronavirus carrier, which can be used to find close contacts of the virus carrier [14,15]. Space-time prisms are usually used to measure the human activity space and the exposure risk of the activity space. This can be used to identify high-risk areas. The projection of the space-time prism on the plane (potential path area, PPA) is generally used to explore the distribution of COVID-19 and to map the potential spatial risk area on the plane. Residents’ daily life, commuting, travel can be planned with these tools of time geography. It can also help countries and regions prevent COVID-19 and map risk areas to assist decision-making [16,17]. Based on the space-time anchor points in the individual’s space-time trajectory, they visualized the space-time uncertainty of the confirmed patient through two key tools of time geography (space-time path and space-time prism). They follow the principle of time geography and infer the unobserved space-time uncertainty based on the observed and confirmed space-time anchor data. This method is effective in describing the space-time uncertainty of individuals in the COVID-19 pandemic.

However, in the context of previous research, the research of time geography has remained at the space-time uncertainty of unobserved locations derived from known observation points. This article will expand in two aspects. First of all, the epidemic is regarded as a polygon feature with moving characteristics, and its core is used as the representative, which is similar to the raster using its center point as the representative. This is different from classical time geography, which uses confirmed individuals with clear boundaries and observables as moving objects. Secondly, we quantify COVID-19 through time geography. COVID-19 is an abstract concept with moving characteristics, including characteristics that change with space and time (such as space-time position, movement speed, etc.). Here we will develop the mathematical basis of how time geography measures the movement characteristics of the epidemic based on statistical data. That is, construct the epidemic space-time disc based on statistical data and then invert the space-time path and the movement characteristics used to measure the epidemic. As a result, time geography can not only describe the space-time uncertainty according to the trajectory, but also describe the trajectory of invisible phenomena based on the statistical data of non-point sources, thus providing quantitative basis for visualization of such phenomena, understanding of historical laws and prediction of future development.

## 2. Time-Geography

Time geography believes that human activities can only occur in specific spatial locations within a limited time [18]. In order to express the location of moving objects in space, time geography provides two key concepts: space-time path and space-time prism [19].

### 2.1. Space-Time Path

The space-time path is actually the life path of a moving object, similar to the concept of lifeline in demographics. As the name implies, the space-time path means that the moving object starts to move from the birthplace (starting point), as time goes by, undergoes many activities in the middle, ends at the death place (endpoint), and stops moving. The same is true for the space-time path of the epidemic. When the first confirmed case appearing in a determinate place, the virus begins to move, and in the mid-term, the virus spreads everywhere. When no new confirmed cases appear in the determinate place, the virus stops moving. However, this is not the space-time path of a virus, but the space-time path of this type of virus.

The space-time path consists of two parts: a set of orderly control points and the corresponding path segments connecting these points. A control point is a position measured in time and space. Control points generally correspond to three types of positions in the real world: known active positions, the position where the path undergoes direction changes (such as turning at a street intersection) or speed change, anywhere where the path records spatial references and timestamps (for example from a GPS receiver) [18].

Path segment is a mathematical theoretical model, and linear interpolation is the traditional method of generating path segments. The path segment generated by linear interpolation represents the direction between the control points, represents the trend of the movement of the moving object, and cannot represent the actual movement trajectory of the moving object. This gives birth to another tool—space-time prism.

### 2.2. Space-Time Prism

The space-time prism in time geography represents the achievable region of a moving object under the constraints of the starting point, ending point, time budget, and maximum velocity, that is, the collection of all possible moving trajectories of the moving object [20]. Mathematically, it means that the sum of the distance from the moving object to the starting point and ending point does not exceed the maximum moving distance of the moving object within a limited time.

According to Figure 1, it is clear that the space-time prism is an irregular cube formed with s as the starting point, e as the endpoint, v as the moving speed, and Δt as the time budget. The space-time prism consists of the intersection of two cones with opposite directions in space. The greater the maximum speed of the moving object, the greater the cone angle of the prism. As is shown in Figure 1, The footprint or projection of the space-time prism in the plane space is called the potential path area [21,22,23]. The potential path area represents the potential space range during the trajectory point pair. In a homogeneous space, it is an ellipse with two trajectory points (s and e) as the focus and ν_max_·(*t_e_* − *t_s_*) as the major axis. Traditionally, space-time path and space-time prism are depicted in a fashion similar to what is illustrated in Figure 1, where the vertical dimension represents time and the horizonal axes represent space.

## 3. Measuring Method of the COVID-19 Based on Time-Geography

In the current paper, we aim to employ the indeterminate statistical data released by the government to retrieve the determinate essential characteristics of COVID-19, and use the information released by the government to verify the COVID-19 spread. The temporal and spatial behavior has been shown to affect livability and wellbeing of individuals. We could understand such effects through the time-geography approach. We deal with the two data through spatial join tool, the temporal and spatial information of COVID-19 reported by the Hebei Provincial Health Commission and the Hebei provincial city-level administrative division map, with the name of the city as the related word. In this way, we obtain a map of administrative divisions with the attributes of COVID-19. Since the temporal and spatial information of COVID-19 cannot specifically describe the detailed activity trajectory of each confirmed patient, and only has the attribute of administrative district, this type of data is defined as indeterminate statistical data. We thereafter select the attributes and location of this special administrative division map to construct the space-time discs at different times and use the epidemic data as the weight to calculate the control points of the corresponding time on the rasterized space-time discs according to the central element method. After that, the control point is connected to simulate the space-time path and the expected trend path. Finally, last but not least, based on the simulated time-space path and the expected trend path, we reversed the parameters of the COVID-19 pandemic, including the starting point, ending point, speed, and development trend. In order to make valid interpretations, we interpret the reversed parameters based on the actual situation and provide experience for COVID-19 prevention and control in other regions.

### 3.1. Transforming COVID-19 Data into Space-Time Discs and Control Points

The space-time discs show all the accessible spatial locations of a moving object at a certain moment [24]. In the homogeneous geographic space, the space-time disc is the intersection of two circles. This intersection may be a circle or an irregular curve. Extending the definition of the space-time disc to the temporal and spatial analysis of COVID-19, the traditional space-time disc has evolved into the area covered by the COVID-19 pandemic at different times.

The specific extraction method of the COVID-19 space-time disc is to combine the time and location of the daily increase of cases with the administrative region and use the geographical name attribute to spatial join the two data so that we get the administrative region with the attribute of COVID-19 information. When there are new cases in certain areas at a certain time, the area involved in these new cases is the space-time disc of COVID-19 at that time. Next, we will simulate the space-time disc of the epidemic through a set of data:Map data: It is the administrative division map of Province A. In order to get better results, we use regular grids as the city area here. In Figure 2, each grid represents an administrative region of a city, which is the smallest spatial unit reported by the Health Commission of Province A.Epidemic data: It includes the number of confirmed cases and the number of newly confirmed cases. The epidemic data were extracted from the Health Commission of Province A.

Here, we simulated 5 days of COVID-19 data. One characteristic of these designed data is that the areas with the largest number of diagnoses vary from one to another every day. In this way, it can be ensured that the points calculated from these data are no longer in the same position, which shows the changes of the new crown epidemic. At the same time, this reflects the mobility of the new crown epidemic. Epidemic space-time disc is a collection of administrative districts with a non-zero increase in the number of cases that day. As is vividly shown, the colored area in Figure 3 represents a COVID-19 space-time disc.

The extraction process of the control point is the process of determining the center point of the space-time disc, that is, the process of determining the point from the surface. There are many ways to determine the center point of polygons. We choose the method of central feature [25]. The calculation formula of the central feature is as follows:(1)$$F=Min(\sum_{i=1}^{n}w_{i}/D_{i})$$

In Equation (1), *i* is any one of the all small grids, *w_i_* is the number of daily new cases of the grid *i*, *D_i_* is the distance from the grid *i* to the point of the central feature, and *n* is the total number of grids *i*.

Central feature identifies the most centrally located feature in all elements through calculating by the sum of the distances from all elements [26].

The attributes of each control point include its location and the time when the location was recorded [18]:(2)$$c_{i}=c(t_{i})=x_{i}$$
(3)$$C=\{(c_{S},\cdot \cdot \cdot ,c_{i},c_{j},\cdot \cdot \cdot ,c_{E})|t_{S}<\cdot \cdot \cdot <t_{i}<t_{j}<\cdot \cdot \cdot t_{E}\}$$

Among them, *c_i_* is the control point, *C* is the aggregate of *c_i_*, *x_i_* is the position of the control point in space, *t_i_* is the time when the control point is recorded, and the set of control points that determine the path is a set of control points sorted strictly according to time. The terms *t_S_* and *t_E_* are the start time and end time of the space-time path, respectively.

### 3.2. Modeling the COVID-19 Space-Time Path 

The path segment is the unobserved position in space that connects adjacent control points in time. The simplest representation of the unobserved position is the straight line segment between the control points. We can use time as a parameter to calculate the interpolation between adjacent control points to define the unobserved path [18]:(4)$$S_{ij}(t)=(1-\alpha )x_{i}+\alpha x_{j}$$
(5)$$\alpha =\frac{t-t_{i}}{t_{j}-t_{i}}$$

Using interpolation to represent paths has both advantages and disadvantages. The advantage of this is that the calculation is simple, convenient, and fast. The disadvantage is that in order to better simulate the real movement trajectory, we need to record a control point at each position where the diffusion velocity (including direction and speed) changes, otherwise it is prone to route distortion. As a result, the number of control points for a simple movement trajectory is much smaller than the number of control points for a complex movement trajectory.

Combining the path segment and the control point using time as the parameter, we can get the expression of the space-time path:(6)$$P(t)=\left\{\begin{array}{l} c_{i},t\in (t_{S},\cdot \cdot \cdot ,t_{i},t_{j},\cdot \cdot \cdot ,t_{E})\\ S_{ij}(t),t_{i}<t<t_{j} \end{array}\right\}$$
where, *S_ij_*(*t*) is the path segment between *t_i_* and *t_j_*.

The space-time path is a polyline formed by calculating the control points of each space-time disc based on the central feature method. The space-time path represents how the center of COVID-19 has changed and developed. Under normal circumstances, we use angle and speed to characterize the space-time path. The angle and speed of the space-time path calculated by projecting the space-time path on the plane is an important basis for quantifying the diffusion velocity and development trend of the COVID-19 pandemic [27]:(7)$$v=\tan\theta =\frac{s}{t}$$
(8)$$v_{A\to B}=\frac{S_{A^{\prime }-B^{\prime }}}{t_{A-B}}$$

$S_{A^{\prime }-B^{\prime }}$ represents the straight-line distance from the control point *A* to control point *B*, and *t_A-B_* represents the time required for the control point *A* to control point *B*. The simulated space-time path is shown in Figure 4.

### 3.3. Modeling the COVID-19 Space-Time Prism 

The space-time dish is a slice of the space-time prism at any moment and it represents the reachable area of a moving object on a two-dimensional plane. The space-time prism is a superposition of space-time discs in chronological order, and is the space-time reachable area of a moving object in three dimensions [28].

Traditional time geography generates a space-time prism from the anchor point (track point) of the moving object, the maximum velocity and the time budget, and then generate the space-time disc from the slice of the space-time prism [29]. Among them, the anchor points are the first and last track points [30]. The most important information required by the anchor point includes the time and space information (*x,y,t*) of the track point. According to the traditional method to generate the space-time prism, the parameter of stay time is also required. Here we have calculated the best stay time after many simulation tests.

The traditional space-time prism is generated in an ideal homogenized geographic space without considering the influence of obstacles. When applying the theory to reality, many unnecessary and unreachable locations are included. Therefore, we need to employ real data to construct a real space-time prism. Our method of constructing the epidemic space-time prism is to form a space-time prism from anchor points and numerous epidemic space-time discs in chronological order. The anchor point is represented as a control point calculated by the central element method on the first space-time disc and the last space-time disc. The anchor point of this article requires the time and space information involved in the first report of the new crown epidemic and the last report of the new crown epidemic in the region. The space-time disc of the epidemic was extracted according to the method described in Section 3.1.

## 4. Implement and Results

### 4.1. Data

The data involved in this article include epidemic data and administrative division maps. The COVID-19 data comes from the epidemic notice issued by the Hebei Provincial Health Commission (https://figshare.com/s/496294de898653f51096) (accessed on 10 August 2020). The research time is from 22 January 2020, to 28 February 2020. In addition to the epidemic data and date, we also employ the city-level administrative division map of Hebei Province from the Geographical National Conditions Detection Cloud Platform. Microsoft Office 2019 (Microsoft, Redmond, WA, USA) was used to process the epidemic data, and ArcMap v. 10.2 (ESRI, Redlands, CA, USA), ArcScene v. 10.2 (ESRI, Redlands, CA, USA) and 3ds Max 2018 (Autodesk, San Rafael, CA, USA) were used for subsequent visualization of the epidemic data. The original epidemic data after preprocessing is shown in Appendix A.

### 4.2. Implement

The extraction process of the space-time disc in Hebei Province is the same as the process simulated above. We apply statistics on the epidemic data of 11 cities in Hebei Province reported by the Health Commission according to time series, and then add the processed statistical data to the attributes of the administrative division map. Finally, the city area where the number of new cases is greater than 0 on the day is selected as the space-time disc of the COVID-19 pandemic. In order to show the 3D display effect, we choose a space-time disc every three or four days. The results of the space-time discs of the epidemic in Hebei Province are shown in Figure 5.

We employ the fishing net tool to divide the administrative division map of Hebei Province obtained from the Geographical National Conditions Detection Cloud Platform according to the size of 10 km × 10 km. The administrative division map of Hebei Province finally consists of 3520 such small squares. We need to divide these small grids into corresponding cities. Since Hebei Province is composed of eleven prefecture-level cities, there will be a mixed grid at the junction of prefecture-level cities, that is, two or more prefecture-level cities appear in one grid. If this happens, it is necessary to determine the ownership of the small square. General methods for determining the value of raster data include the center attribution method, area dominance method, length dominance method, and importance method [31]. What we choose is the area dominance method, that is, the area of city a in the mixed grid is larger than the area of city b, then the mixed grid belongs to city a, but if special circumstance happens, i.e., city c is completely contained by the mixed grid, regardless of whether the area of city c exceeds city a or city b, then the mixed grid belongs to city c, which is a judgment based on the importance method.

Figure 6 shows the result of rasterizing the administrative regions of Hebei Province according to the area-dominant method.

According to the above-mentioned space-time discs extraction method, we draw the rasterized space-time discs. On this basis, we extracted the control points of the COVID-19 space-time discs at different moments according to the central feature method as shown in Figure 7.

The parameters of the time-space path calculated based on the actual epidemic data in Hebei Province are shown in Table 1.

The two-dimensional and three-dimensional time-space paths drawn based on the real epidemic data of Hebei Province are shown in Figure 8. In the left image of Figure 8, the green dot represents the starting point. The yellow point represents the end point, and the blue point represents the central feature point. The red arrowed line represents the two-dimensional space-time path.

As is shown in the Figure 9, we can obtain the parameters of the maximum diffusion velocity and stay time of the COVID-19 pandemic by analyzing the simulated epidemic space-time prism according to our method. The maximum diffusion velocity is 142.489 km/h and the stay time is 17 days. The maximum diffusion velocity represents the spread of the infection. Because we have not studied the spread of COVID-19 in other provinces, we cannot compare the spread of COVID-19 in Hebei with that of other provinces. However, through the length and width of Hebei Province calculated by this projection and the maximum diffusion velocity, we can conclude that if the COVID-19 pandemic spreads at the maximum speed, COVID-19 would cover the entire Hebei Province within 7 days. The stay time starts on 28 January 2020 and ends on 13 February 2020. The stay time represents that the new crown epidemic lasted for 17 days in Hebei Province at the maximum spread speed. It took 7 days from the first case in Hebei Province on 22 January 2020 to the new crown epidemic covering the whole province on 28 January 2020. This happens to be consistent with the conclusion obtained above.

In summary, we have established the space-time path and space-time prism of the COVID-19 pandemic by examining the opposite method, compared with the traditional time-geographic modeling method. At the same time, we extracted the maximum diffusion velocity parameters and stay time parameters of COVID-19. With this we have achieved our expected goal.

### 4.3. Results

With the above-mentioned time geography methods, we can measure the COVID-19 pandemic spread, including starting point, ending point, expected trend path and diffusion velocity. The starting point, the ending point and the expected trend path are shown in Figure 8. The parameters of the diffusion velocity are shown in Table 1.

The starting point is composed of two parameters: the position and time of the control point. The time is the date corresponding to the first space-time disc: 22 January 2020, and the location is the administrative district corresponding to the first control point: Shijiazhuang. The end time is the date corresponding to the last space-time disc: 19 February 2020, and the location is the administrative district corresponding to the last control point: Tangshan. The set of line segments connected by the control points on each space-time disc in chronological order is the actual trajectory of the center of pandemic. The expected trend path is the connection between the two anchor points, that is, the connection between the first and last control points, and does not involve the control point in the middle position. All the control points are connected in chronological order to form a polyline that represents the actual movement trajectory of the center of COVID-19, and the expected trend path represents the trend of the movement of the epidemic. The expected trend path is the purple solid line in Figure 10, which represents that the COVID-19 pandemic in Hebei Province has gradually spread from Shijiazhuang as the center to Tangshan as the center, at an angle of 64.93 degrees east of north. The red polyline in the Figure 8 is the space-time path of COVID-19. The overall expected trend is from north to south and west to east. It is expected that the results presented by the expected trend path and the space-time path are consistent with the results of the control point analysis. Diffusion velocity represents the direction and scope of the spread of the center of COVID-19. According to the distribution of angles, the distribution center of the COVID-19 pandemic presents a distribution from east to west in the east-west direction, and the north-south upwards presents the fluctuating distribution around the expected trend path. In general, it moves from south to north. The results of the speed show that the distribution center of the COVID-19 pandemic is moving at an uneven speed, but the whole process has the characteristics of increasing-stable-decreasing. This feature is similar to that of general infectious diseases, that is, rapid spread in the early stage, stable in the middle stage, and contraction in the late stage. From the calculated speed of each space-time disc, it can be seen that the moving object of COVID-19 is unstable and highly random.

Among the nine control points calculated by the selected space-time discs, three of them are located in Shijiazhuang City, two are located in Langfang, two are located in Tangshan, and the remaining two are located in Zhangjiakou and Cangzhou. The center feature method is different from the average center method and other methods for calculating the center point of the surface. The central feature method is to select a point in the existing points set as the central feature, and the average center method is to generate a new point as the average center based on the existing points set. Therefore, the advantage of determining the control point by the central feature method is reflected: it will not choose an area without an epidemic as the center of the COVID-19 pandemic.

These nine control points involve five cities. Considering these nine control points and the number of newly confirmed in these eleven cities in Hebei Province, the following conclusions can be drawn. These five cities are the centers of different development levels of the COVID-19 pandemic. The first level, the initial level of development of COVID-19, from 22 January 2020 to 2 February 2020, starting from Shijiazhuang, while Shijiazhuang is the center, and Zhangjiakou is the secondary center. COVID-19 gradually spread to the whole province; this could be related to the migration during the Spring Festival travel season in 2020, according to the Baidu Migration Index, Shijiazhuang ranks the highest in the province of the number of people who migrated from Wuhan to Hebei Province during the Spring Festival travel season [32]. At the same time, transportation in Hebei Province is developed. According to the strength of provincial spatial transportation connections, Shijiazhuang City is the main core of road transportation in Hebei Province [33]. In the second phase, from 3 February 2020 to 16 February 2020, with Langfang and Cangzhou as the center, the development of the COVID-19 has reached its climax. Hebei Province launched the first-level response to major public health emergencies in 25 January 2020, which means that the second level of Hebei Province is mainly the population movement within the province. It also means that in this phase, the newly confirmed cases mainly came from mutual infections within the province. Because Shijiazhuang was the starting point of the COVID-19 pandemic in Hebei Province, and at the same time the main core of road transportation in the province, this also exacerbates the rapid development of the epidemic in the second phase. The GDP and population of the cities in the north-central Hebei province are higher than those in the south, which means that the economic activities and population movements in Hebei are mainly concentrated in the north and center [34]. The development of the epidemic requires the flow of population. Therefore, the center of the epidemic has moved from the southern part of the beginning to the central cities. Coupled with the fact that Langfang and Cangzhou are located in the geographic center of Hebei Province, the cumulative number of confirmed cases in Langfang and Cangzhou in the second phase is more than that in other cities, which has led to the second phase of the epidemic center in Langfang and Cangzhou. In the third phase, from 17 February 2020 to 28 February 2020, the epidemic has entered its final phase. The number of newly confirmed patients has dropped rapidly. The development of the pandemic in some cities has stagnated, and no new confirmed cases will appear. At the same time, with Tangshan as the center, sporadic newly confirmed cases appear in individual cities, and they are in a repeated state.

### 4.4. Verification

The standard deviation ellipse can be used to quantify this trend of distribution [35]. The major semi-axis of the standard deviation ellipse represents the direction of the data distribution, and the minor semi-axis represents the range of the data distribution [36]. The larger the difference between the long and short semi-axis values (the greater the flatness), the more obvious the directionality of the data. Conversely, if the long and short semi-axes are closer, the directivity is less obvious. If the long and short semi-axes are exactly the same, it is equal to a circle. A circle means that there is no directional feature [37]. The semi-minor axis represents the range of data distribution. The shorter the semi-minor axis, the more obvious the centripetal force presented by the data. Conversely, the longer the semi-minor axis, the greater the degree of dispersion of the data. Similarly, if the semi-minor axis is exactly equal to the semi-major axis, it means that the data does not have any distribution characteristics.

The standard deviation ellipse calculated with the nine control points is the blue ellipse in Figure 10. The parameters of the standard deviation ellipse are shown in Table 2. The result obtained by the standard deviation ellipse is that the distribution center of the COVID-19 infection in Hebei Province is Langfang, the distribution of the COVID-19 pandemic is in the southwest-northeast direction, and the deflection angle is 63.034 degrees north to east. The difference between the direction distribution value calculated by using the standard deviation ellipse and the direction distribution value calculated by the expected trend path is very small, so the standard deviation ellipse can be used to verify the calculated space-time path parameters.

Location entropy ($E_{p}=-\sum_{i=1}^{K}p_{i}\log_{2}p_{i}$) can be used to characterize the randomness of moving objects visiting different places [38,39,40]. In this expression *p_i_* is the probability of a moving object visiting location *i* and *K* is the total number of different locations visited by a moving object.

The lower the entropy value, the more fixed the place the moving objects visiting. Conversely, the higher the entropy value, the more random the place the moving objects visiting. It can be seen from Table 3 that, at the beginning and end of the epidemic, the epidemic spreads in a small area, the entropy value is small, the randomness is small, and the position of the epidemic is relatively fixed. In the middle of the rapid development of the epidemic, the epidemic spreads in a large area, the entropy value is large, the randomness is large, and the location of the epidemic is easy to change. In addition, the correlation coefficient between position entropy and area of space-time disc is 0.928, and the two have a strong correlation. At the same time, the overall position entropy shows a law of increasing first, stable in the middle and decreasing later, which is consistent with the law embodied by diffusion velocity. Therefore, the position entropy can be used to verify the result calculated by diffusion velocity.

## 5. Discussion

In this paper, our idea is to show the geographic mobility patterns of the COVID-19 pandemic within a province based on the provincial space-time prism. However, that also exposes the limitations of the proposed method, that is, the theoretical space-time prism is not affected by the provincial boundary, but rather a region that covers different provinces. The space-time prism has a clear boundary, and its boundary will conflict with the boundary of the study area [41,42,43]. Therefore, in the next step, we will study the spread of the epidemic from a multi-regional perspective, such as replacing a single provincial boundary with multiple provinces as a whole, so as to better restore the spread of the epidemic in a large-scale area. Our current method (employing the province as the boundary) limits the spread of the epidemic from the region to the province. This method is suitable for studying the epidemic in the province. For example, in 2020, all provinces adopt home isolation measures. There is basically no large population flow across provinces, and the spread of the epidemic is restricted within the province. Therefore, the current space-time prism based on the provincial boundary has a certain realistic foundation. The method is effective for the 2020 pandemic which spread widely and involved a large number of people. As for the current scattered distribution of Delta virus in China, when the provinces involve a small population, it is effective to adopt inter-provincial regional boundaries.

Different spatial scales will have a great impact on the research results [44,45,46]. This is also explained in the Introduction section. On a macro scale, our research can only reflect the transmission characteristics of the COVID-19 pandemic at the city level, and cannot simulate the actual natural transmission chain of COVID-19. At the same time, since the smallest research unit we chose is the city, which is composed of many natural villages, counties, and districts, we will equally distribute all the COVID-19 data in the city to all natural villages, districts, and counties, which will include many areas where COVID-19 has not occurred. This will cause errors. In fact, there is a widespread scale effect problem in geography. The results at the natural village scale are definitely more accurate than those at the city scale. However, a solution of the problem may also bring new problems. Analysis of the natural village scale requires a larger amount of data and the constructed model is more complex. Moreover, natural villages can also be divided into residential areas and natural areas. This will bring a new round of scale effect problem. In this paper, the idea is to construct a new time geography model through the time geography framework to analyze and visualize the characteristic of the COVID-19 pandemic in Hebei Province, and verifies the accuracy of the model we constructed through methods such as standard deviation ellipse and location entropy.

## 6. Conclusions

Based on the time-geography framework and the COVID-19 data, in this paper, the idea is to construct a model that is different from the classical time geography one. The classical time geography model needs some parameters such as the starting point, the ending point, moving speed, the time budget and a series of control points in the middle of a moving object, so that we can draw a space-time path representing the movement trend and a space-time prism representing the temporal and spatial achievable region. In general, the classical time geography model analyzes indeterminate movement trends and reachable areas based on deterministic data. The model proposed in this paper first outlines the temporal and spatial achievable region from a large amount of fuzzy statistical data, thereafter simulates the space-time prism according to the temporal and spatial achievable region, generates control points according to the space-time disc slicing by the space-time prism, and then simulates the space-time path. Last but not least, we can reverse some essential characteristics of the COVID-19 pandemic (stay time, diffusion velocity, movement route, etc.). It can be seen from this that an overall outline for our idea is to analyze and visualize the characteristics based on the vague data set. This is not the same as the classical time geography model.

The results indicate that the COVID-19 pandemic in Hebei Province shows a trend of spreading from the southwest to the northeast on the whole, and in the early stages of the epidemic, the diffusion velocity of COVID-19 is rapid. However, due to strict restrictions (restrictions on travel, restrictions on shopping, etc.), the development of COVID-19 has been greatly circumscribed.

In the future, we will re-analyze the COVID-19 data in Hebei Province at the natural village scale to further verify the accuracy of the model. Analyzing and visualizing the characteristics of the COVID-19 pandemic are relevant to the current pressing need and have realistic significance, which can help people get out of trouble as soon as possible and return the world to order as soon as possible. The WHO could consider this model to play a significant role in exploring the temporal and spatial distribution, making health decisions, and formulating strategies. This study may provide scientific and practical reference on the prevention and control of COVID-19 for decision makers.

## Figures and Tables

**Figure 1 ijerph-18-10313-f001:**
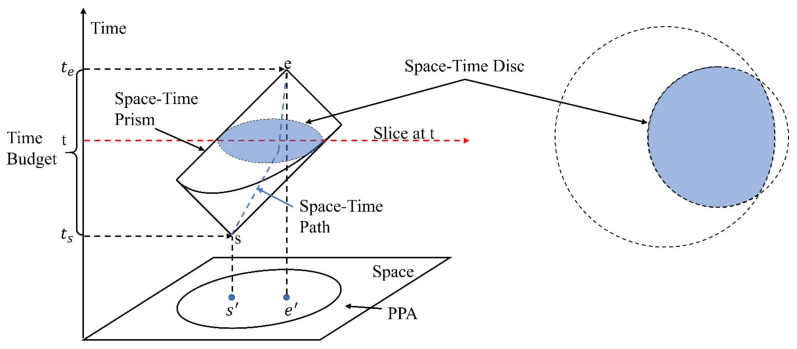
Parameters in the space-time prism.

**Figure 2 ijerph-18-10313-f002:**
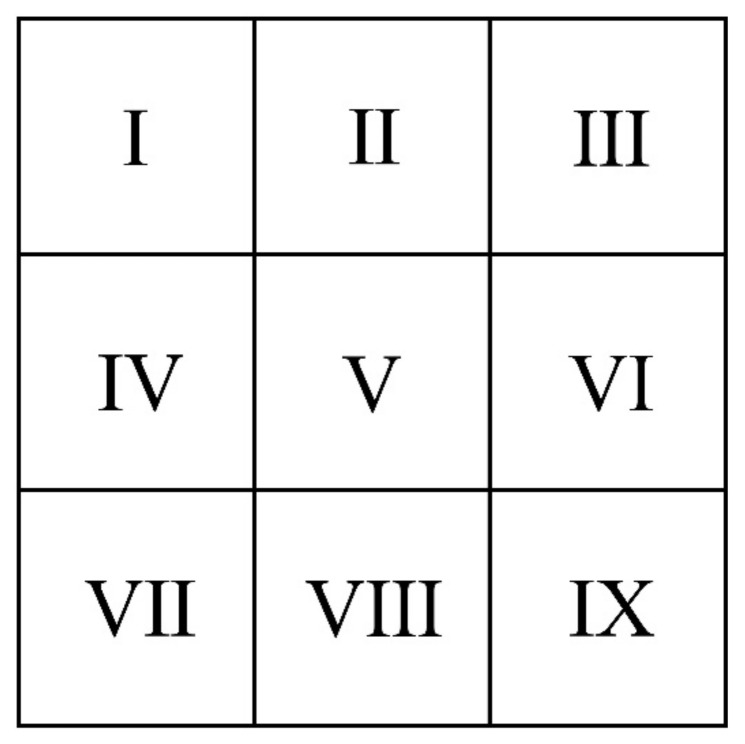
Administrative area of Province A.

**Figure 3 ijerph-18-10313-f003:**
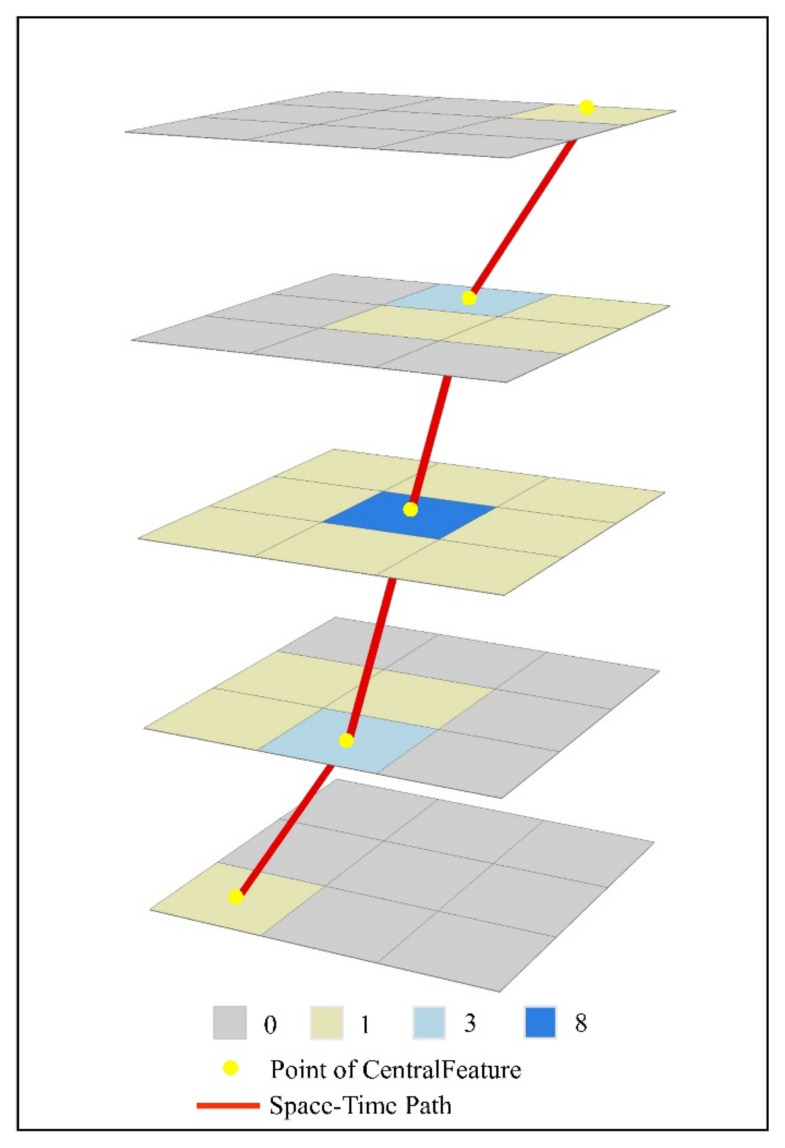
Simulated space-time discs.

**Figure 4 ijerph-18-10313-f004:**
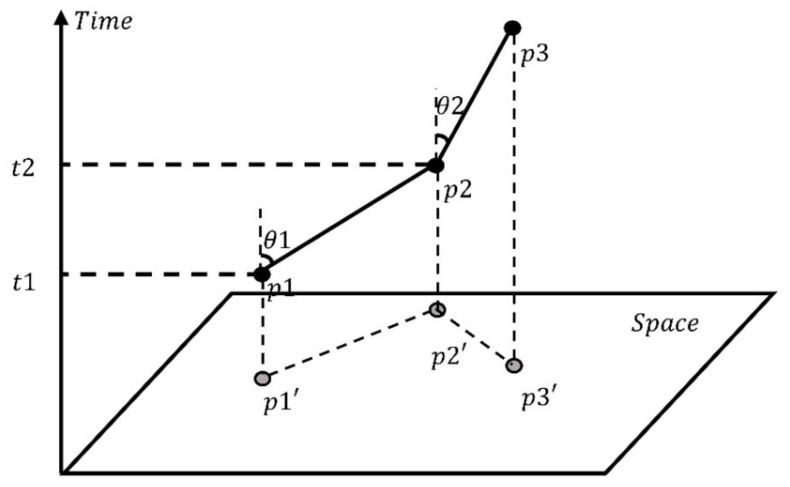
A simulated space-time path.

**Figure 5 ijerph-18-10313-f005:**
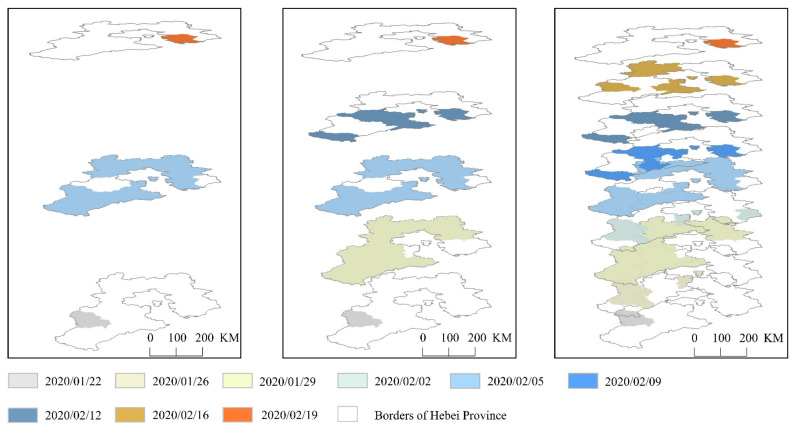
Hebei space-time discs.

**Figure 6 ijerph-18-10313-f006:**
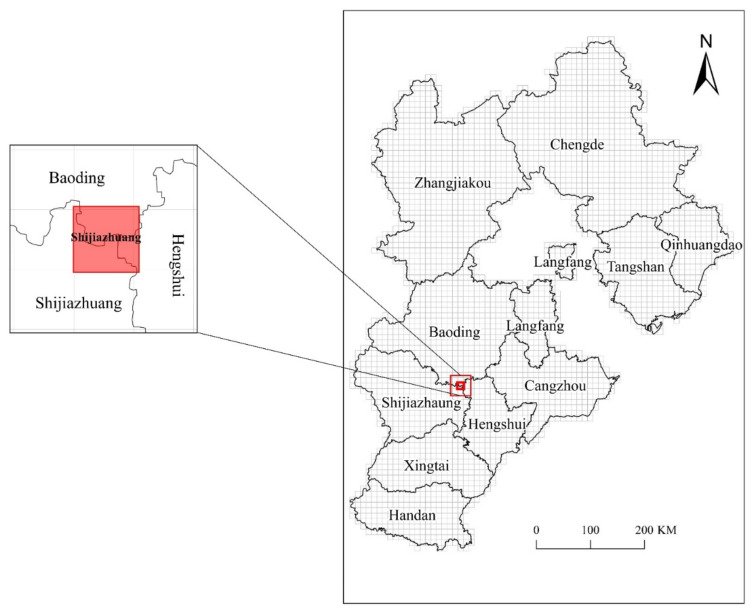
Rasterized Hebei Province.

**Figure 7 ijerph-18-10313-f007:**
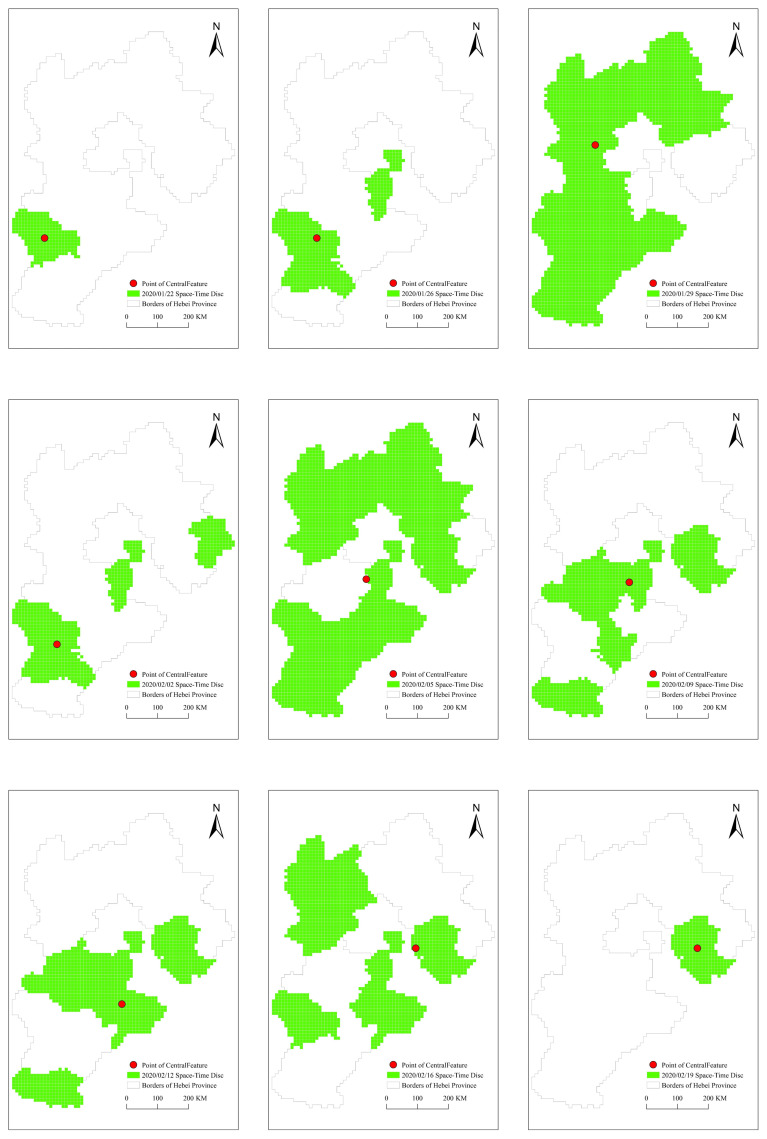
Control points in the space-time discs.

**Figure 8 ijerph-18-10313-f008:**
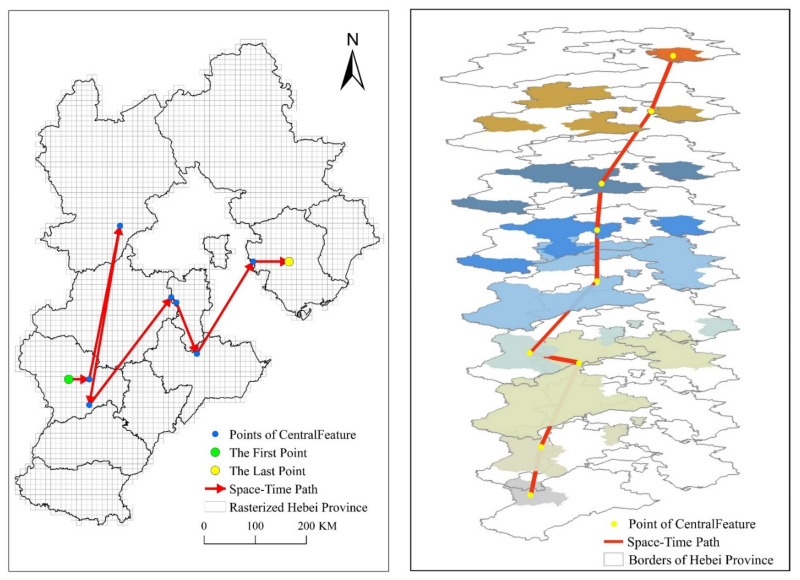
Space-time paths (2D (**left**), 3D (**right**)).

**Figure 9 ijerph-18-10313-f009:**
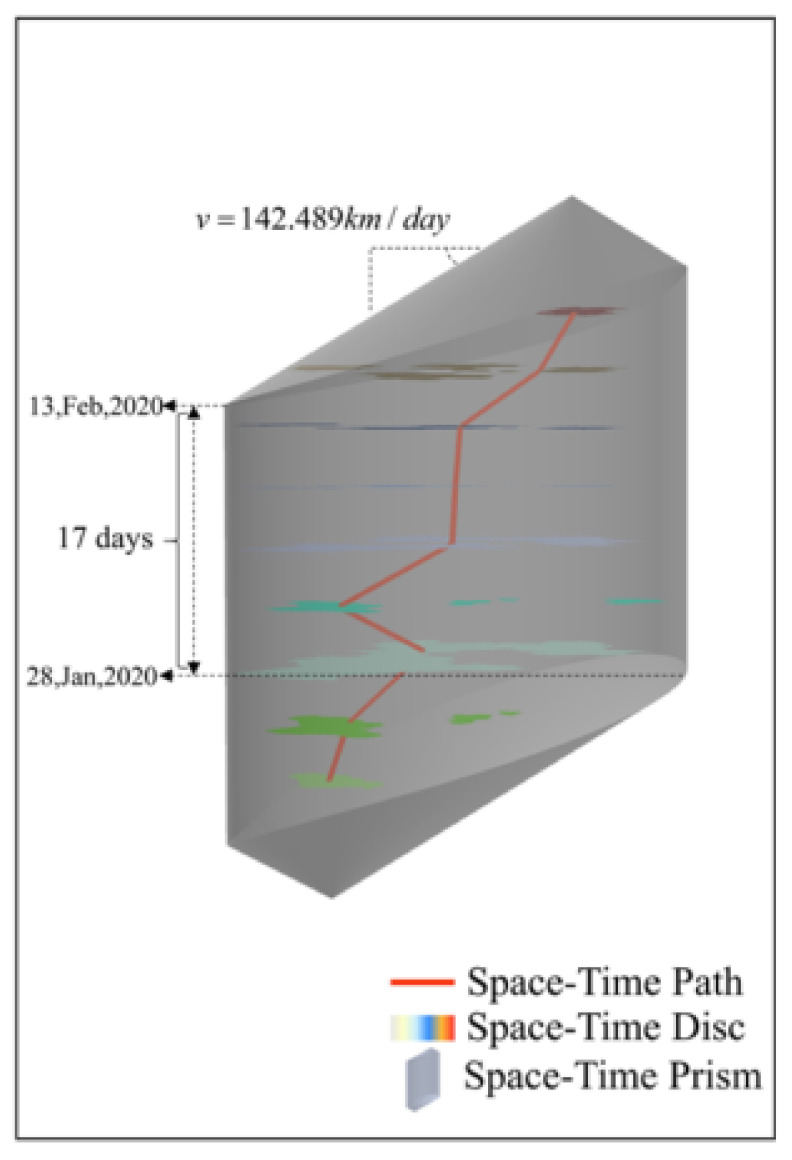
The space-time prism of Hebei Province.

**Figure 10 ijerph-18-10313-f010:**
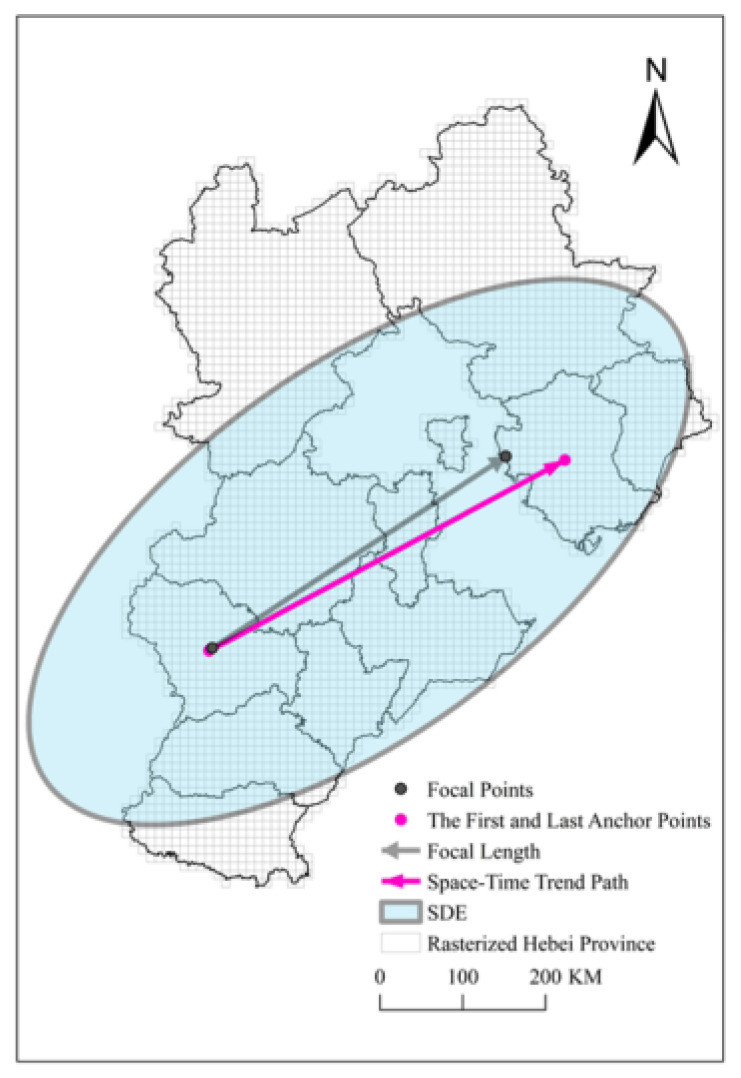
The standard deviation ellipse and expected trend path.

**Table 1 ijerph-18-10313-t001:** The parameters of the space-time path on the two-dimensional plane.

* Parameters	Jan. 22–Jan. 26	Jan. 26–Jan. 29	Jan. 29–Feb. 02	Feb. 02–Feb. 05	Feb. 05–Feb. 09	Feb. 09–Feb. 12	Feb. 12–Feb. 16	Feb. 16–Feb. 19
θ (°)	5.133	35.833	32.133	34.541	1.630	14.742	21.819	11.838
Direction	east	north by east	south by west	north by east	south by east	south by east	north by east	east
Speed(km/days)	10.000	101.98	88.776	88.003	3.536	35.901	52.738	23.333

* Notes: The space-time path has two very important attributes: the velocity and angle (θ) between two control points. velocity is composed of speed and direction. So, we can use θ, velocity and direction to describe the space-time path.

**Table 2 ijerph-18-10313-t002:** The parameters of the standard deviation ellipse.

Parameters	CenterX	CenterY	XStdDist	YStdDist	Rotation
Value	116.025	38.934	1.642	3.934	63.034

**Table 3 ijerph-18-10313-t003:** Location entropy.

Date	2021/01/22	2021/01/26	2021/01/29	2021/02/02	2021/02/05	2021/02/09	2021/02/12	2021/02/16	2021/02/19
Location entropy	7.966	9.084	11.155	9.408	10.968	10.050	10.004	9.983	8.066
Area ((100m)^2^)	250	590	2961	743	2975	1159	1277	1636	268

## Data Availability

Raw data with detailed description is available online: https://figshare.com/s/496294de898653f51096 (accessed on 10 August 2020).

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
