# Peer review of "Measuring of the COVID-19 Based on Time-Geography"

_ijerph, 2021, doi:10.3390/ijerph181910313_

Round 1
Reviewer 1 Report
29-30 it doesen’t sound
151-152 ??
Table 1 is it an example? Why not to try to give the whole database, maybe as annex. Align better the notes
191-192 ??
205-215 It clear but the figure can be postponed and the description needs more fluidity. After “data” use : and not .
Improve the quality of figures
Figure 3 a/b make two different figures
In general, some explications present in the text can be shown in the captions
Figure 4 the caption is far from the figure
I missed a point: How do you consider in you method the way to control the number of the COVID affected? Is it the same to have 1 or 10 or 100 COVID affected?
Figure 5 it is difficult to follow. Should it have a kind of parallelism with figure 3 or not? It doesn’t seem it.
Table 2 it is not clear
Figure 7 the 2Dpath is difficult to follow, while the 3D one explain very well the meaning of the paper
315-316 It seems a tongue twister
338 Maybe “it” is missed, and I shouldn’t use the term vividly
342 Maybe to have some data to compare could help
352 successfully??? You established it, but you don’t have a term of comparison
375 yellow??
Reviewer 2 Report
Presented manuscript concentrated on the reveal spatial and temporal pattern of the COVID-19 in Hebei Provinceby using time-geography tools, such as space-time path and space-time prism. The topic and the main ideas of the article are relevant for the International Journal of Environmental Research and Public Health.
I really appreciate the article, the issues, and the question posed and analyzed by the authors, methods and tools. The topic is very interesting, actual and innovative. However, I have the following comments that are essential, but I believe that once the Authors have incorporated them, the manuscript could be accepted for publication:
(1) The literature review part should be strengthened (line 94-100), to clearly summarize what have been found while what have not in order to demonstrate the contribution of this case study.
(2) The methodology is well developed and referenced but the article is not well constructed. The methodology section contains partially obtained and discussed results, which should be included in the Chapter 3. Results. The chapter on methodology is not easy to read as it contains partial results. In the Section 2 - Materials and Methods, the authors should reorganise the text.
(3) The same with the chapter on the results obtained. There, too, are the methodological assumptions, which should be described in the methodology Chapter, e.g. formula 9.
(4) I suggest reviewing a little bit the Discussion, Chapter 4 is not a scientific discussion. This chapter is of a concluding nature and includes assumptions for future research. There is a lack of reference of the results obtained to other researchers, which is important in scientific publication.
(5) Chapter 2.1, can be considered and titled as assumptions/theoretical foundations, especially as the authors in chapter 2.1.1. and 2.1.2. describe two key concept.
(6) In chapter 2.1.1. there is no reference in the text to Figure 1. Figure 1 should be moved to chapter 2.1.2.
(7) Same repetition in lines 149-153 and 189-193, please correct. Is this text important?, the paper identifies sources of data.
(8) In line 262 there is a title for figure 4, this figure should be moved.
(9) In Figure 5, invisible legend, please correct.
(10) There are a few editorial errors e.g. line 313, no space in the title of Figure 7. In Figure 7 it would be good to distinguish the end point from the start point, although this can be guessed by analysing the data in table 2, but the strength of any map and legend is that we look at it and have no doubt…
(11) Table 2. The parameters of the space-time path (line 306-308) and Figure 7 (line 310-311) are already results and are provided in the methodology section.
(12) Line 337-338 „The space-time disc of the epidemic was extracted according to the method in 3.1.2” there should probably be a reference to section 2.1.2?
(13) Part of the methodology section (chapter 2.2.5) are the results already obtained, e.g. line 344-356.
(14) In line 361 there is a reference to Figure 9, which is only after line 448. The reference to Table 2 in line 361 indicates that Table 2 is the result although it is in the Methodology Section.
(15) In lines 538-541 there is no indication for which specific administrations, this methodology is recommended, whether it can e.g. improve crisis management plans during such epidemics, care priorities, etc. This would increase the value of the manuscript.
